# Success or Failure of Chiral Crystallization of Similar Heterocyclic Compounds

**DOI:** 10.3390/molecules25235691

**Published:** 2020-12-02

**Authors:** Cyprian M. Chunkang, Iris E. Ikome, Emmanuel N. Nfor, Yuta Mitani, Natsuki Katsuumi, Tomoyuki Haraguchi, Takashiro Akitsu

**Affiliations:** 1Department of Chemistry, Faculty of Science, University of Buea, Buea P.O. Box 63, Cameroon; mchunkang@yahoo.com (C.M.C.); chili_iris@yahoo.co.uk (I.E.I.); nfor.emmanuel@ubuea.cm (E.N.N.); 2Department of Chemistry, Faculty of Science, Tokyo University of Science, 1-3 Kagurazaka, Shinjuku-ku, Tokyo 162-8601, Japan; 1320605@ed.tus.ac.jp (Y.M.); 1320529@ed.tus.ac.jp (N.K.); haraguchi@rs.tus.ac.jp (T.H.)

**Keywords:** heterocycle compound, chiral crystallization, crystal structure, space group, hydrogen bonds

## Abstract

Single crystals of two achiral and planar heterocyclic compounds, C_9_H_8_H_3_O(**CA1**) and C_8_H_5_NO_2_ (**CA4**), recrystallized from ethanol, were characterized by single crystal X-ray analysis, respectively, and chiral crystallization was observed only for **CA1** as *P*2_1_2_1_2_1_ (# 19), whereas it was not observed for **CA4**
*P*2_1_/c (# 14). In **CA1**, as a monohydrate, the hydrogen bonds were pronounced around the water of crystallization (O4), and the planar cyclic sites were arranged in parallel to slightly tilted positions. On the other hand, an anhydride **CA4** formed a dimer by hydrogen bonds between adjacent molecules in the crystal, which were aggregated by van der Waals forces and placed in parallel planar cyclic sites.

## 1. Introduction

4-Hydroxycoumarins (2H-1-benzopyran-2-ones) have evoked a great deal of interest due to their biological properties and characteristic conjugated molecular architecture. Many of them have been found to display interesting pharmacological effects, including analgesic [1]. They have been effectively used as anticoagulants for the treatment of disorders in which there is excessive or undesirable clotting, such as thrombophlebitis [2], pulmonary embolism [3], and certain cardiac conditions [4]. A number of comparative pharmacological investigations of the 4-hydroxycoumarin derivatives have shown good anticoagulant activity combined with low side effects and little toxicity [5]. The increasing pharmacological use of these compounds is also attributed to their tendency to form complexes with biologically important transition metal ions.

Isatin (1H-indole-2,3-dione) and its derivatives, on the other hand, represent an important class of heterocyclic compounds that can be used as precursors for drug synthesis. They have been reported to have a wide range of pharmacological activities, including antiviral [6], spermicidal [7], anti-corrosive [8], analgesic [9], anticonvulsant [10], antioxidant [11], antitubercular [12], transthyretin fibrillogenesis inhibitory [13], antidepressant [14], and antiepileptic [15] activity.

On the other hand, chirality is a useful concept in physical and life sciences, especially when applied to a molecular level. Chirality manifests itself in both molecules and crystals, and its origin lies clearly in molecular architecture. The most common structural motif encountered in chiral molecules is the chiral center, usually a carbon atom surrounded by four different substituents (Ca_bcd_). Chirality in molecules devoid of chiral centers occurs in both natural and synthetic substances, which have enormous importance as ligands in asymmetric catalysis, with some exhibiting biological activity [16]. The chirality of drugs has become a significant topic in the discovery, design, patenting, and marketing of new pharmaceuticals [17,18,19]. This has led to the increasing realization of the significance of the pharmacodynamic and pharmacokinetic difference between enantiomers of chiral drugs in therapeutics.

Furthermore, beyond conventional “molecular chirality” associated with pharmaceutical compounds, “supramolecular chirality” attributed to molecular aggregation may be important to understand the origin of chirality by weak forces. Especially, when achiral molecules form chiral crystals, in other word, crystallize in the chiral space group, what factors should be required for molecule or intermolecular interactions? To know such factors, comparison between chiral and achiral crystals of similar compounds may be useful.

In this study, chiral crystallization (achiral compounds crystallize in the chiral space group) [20,21,22] was observed in a heterocyclic 4-hydroxycoumarin compound (**CA1**), accidentally obtained in trying to synthesize a hydrazone Schiff base ligand, by reacting 4-hydroxycoumarin and pyrazinamide in the presence of glacial acetic acid in ethanolic solution. However, chiral crystallization was not observed in a heterocyclic compound (**CA4**), accidentally obtained in trying to synthesize a Schiff base ligand, by reacting isatin and pyrazinamide in the same procedure as for **CA1**. This study, therefore, aimed to obtain the structural information necessary for discussing “supramolecular chirality” of achiral compounds with potential application for the pharmacological properties of the synthesized compounds.

## 2. Experimental

### 2.1. Synthesis and Crystallization of CA1

The compound **CA1** was prepared by the reaction of 4-hydroxycoumarin (1.50 g, 0.01 mol) and pyrazinamide (1.14 g, 0.01 mol) in 25 mL of ethanolic solution with three drops of glacial acetate added as a catalyst. The mixture was then refluxed at 90 °C while stirring for 5 h. The product was left overnight to cool; removed by vacuum filtration; washed several times with water, ethanol, and diethyl ether; and finally crystallized from ethanol after 30 days as colorless crystals suitable for single crystal X-ray diffraction studies; yield 70%. The crude products of **CA1** and **CA4** (in the next section) before recrystallization for purification and growing single crystals were obtained directly from the reaction solutions as the sole but unexpected product.

### 2.2. Synthesis and Crystallization of CA4

The compound **CA4** was prepared by the reaction of isatin (1.50 g, 0.01 mol) and pyrazinamide (1.26 g, 0.01 mol) in 25 mL of ethanolic solution with three drops of glacial acetate added as a catalyst. The mixture was then refluxed at 70 °C while stirring for 5 h. The product was left overnight to cool; removed by vacuum filtration, washed several times with water, ethanol, and diethyl ether; and finally crystallized from ethanol after two weeks as orange crystals suitable for single crystal X-ray diffraction studies; yield 80%.

### 2.3. Analytical Methods

Infrared (IR) spectra were measured on a JASCO (Tokyo, Japan) FT-IR 4200 spectrophotometer in the range of 400–4000 cm^−1^ at 298 K. UV-vis (electronic) spectra were measured on a JASCO (Tokyo, Japan) V-650 spectrophotometer in the range of 800–220 nm at 298 K.^1^H-NMR spectra were recorded on a JEOL JMN-300 spectrometer (300 MHz) (JEOL, Tokyo, Japan). Due to the small amount of sample obtained by accident, elemental analysis could not be carried out.

### 2.4. X-ray Crystallography

A colorless (**CA1**) or an orange (**CA4**) prism crystal having approximated dimensions of 0.200 × 0.100 × 0.200 (**CA1**) and 0.120 × 0.100 × 0.020 mm (**CA4**) was mounted on a glass fiber. All measurements were made at 203 K on a Rigaku R-AXIS RAPID diffractometer using multi-layer mirror monochromated Mo-Kα radiation (λ = 0.071069 nm).

The structure was solved by direct methods [23] and expanded using Fourier techniques. The non-hydrogen atoms were refined anisotropically. Hydrogen atoms were refined using the riding model. The final cycle of full-matrix least-squares refinement [24] on *F*^2^ was based on 1883 and 1440 observed reflections and 127 and 100 variable parameters and converged (largest parameter shift was 0.00 times its esd) with an unweighted and weighted agreement factor. All calculations were performed using the CrystalStructure [25] crystallographic software package except for refinement, which was performed using SHELXL Version 2018/3 [23]. Crystallographic data are summarized in Table 1 and available from CCDC (see Appendix A).

### 2.5. DFT Computation

The density functional theory (DFT) calculations of the optimized structure of **CA1** and **CA4** were carried out using the Gaussian 09W software package Revision D.02 (Gaussian, Inc., Wallingford, CT, USA) [26] with a Windows 10 personal computer. All geometries were optimized by using the B3LYP level of theory and basis set SDD. The frequency calculations were performed for the optimization of geometries (Figure 1) using the identical level of theory and basis set.

## 3. Results and Discussion

Prior to structural discussion, it was confirmed that IR, UV-vis, and ^1^H-NMR spectra were in agreement with the simulated ones by DFT computation and assigned reasonably (Figure 2). Expected NMR peaks appear at a = 7.39, b = 7.70, c = 7.40, and d = 7.84 ppm for **CA1** and a = 7.81, b = 7.68, c = 7.32, and d = 8.02 ppm for **CA4**.

Both molecules adopted planar structures, with all the heavy atoms on one mean plane. As for **CA1**, the six-membered carbon ring and heterocyclic moiety in the compound adopt a planar structure orientation, and are arranged in a parallel to slightly tilted position in the crystal. Significant hydrogen-bonding interactions of O1-H1A⋯O4 (O…O = 2.5578(13) Å), O4-H4A⋯O3 (O…O = 2.7659(15) Å), and O4-H4B⋯O3′ (O…O = 2.7443(15) Å) are very much evident in the compound (Figure 3) (symmetry operation ‘: 1/2 − x, 1 − y, −1/2 + z).

As for **CA4**, the six-membered carbon ring and heterocyclic moiety in the compound adopt a planar structure orientation, and is parallelly arranged in the crystal. Significant hydrogen bonding interactions of N1-H1⋯O2 (O…O = 2.9017(14) Å) are very much evident in the compound (Figure 4).

Comparing the crystal packing of **CA1** and **CA4** (Figure 5), in the chiral crystallized **CA1**, the hydrogen bonds are pronounced around the water of crystallization (O4), and the planar cyclic sites are arranged in parallel to slightly tilted positions. On the other hand, the non-chiral crystallized **CA4** did not contain crystallized water, and the hydrogen bonds between adjacent molecules formed a dimer, which is aggregated by van der Waals forces and results in a parallel arrangement of planar cyclic sites. These results suggest that one of the reasons for the chiral crystallization in **CA1** is that the alignment of the planar cyclic sites was slightly tilted from parallel in the crystal packing. Similar intermolecular interaction may also appear in the discussion of helical chirality. As for the sterically driven mechanism for the formation of supramolecular helicity in the solid state, sterically bulky groups of spiral arrangement, uninterrupted hydrogen-bonding chains, and repulsive stereochemical interactions are required for serving as an axis for the helical structure [27].

## 4. Conclusions

In this study, some factors of “supramolecular chirality” of achiral compounds were described in comparison with similar achiral compounds as suitable examples of chiral crystallization. Single crystals of two achiralplanar heterocyclic compounds, **CA1** and **CA4**, recrystallized from ethanol, were analyzed by structural analysis, respectively, and chiral crystallization was observed at chiral *P*2_1_2_1_2_1_ (# 19) in **CA1**, whereas it was not observed at achiral *P*2_1_/c (# 14) in **CA4**. Comparing the crystal packing of **CA1** and **CA4** (Figure 4 and Figure 5), in the chiral crystallized **CA1**, the hydrogen bonds were pronounced around the water of crystallization (O4), and the planar cyclic sites were arranged in parallel to slightly tilted positions. On the other hand, the non-chiral crystallized **CA4** did not contain crystallized water, and the hydrogen bonds between adjacent molecules formed dimer, which is aggregated by van der Waals forces and results in a parallel arrangement of planar cyclic sites. These results suggest that one of the reasons for the chiral crystallization in **CA1** is that the alignment of the planar cyclic sites was slightly tilted from parallel in the crystal packing.

## Figures and Tables

**Figure 1 molecules-25-05691-f001:**
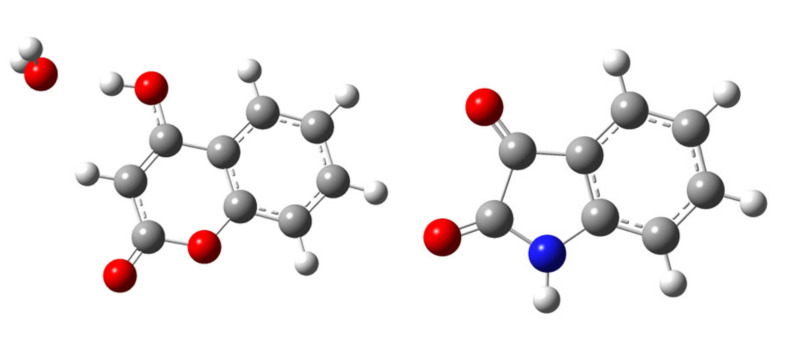
DFT optimized molecular structures of (**left**) **CA1** and (**right**) **CA4**.

**Figure 2 molecules-25-05691-f002:**
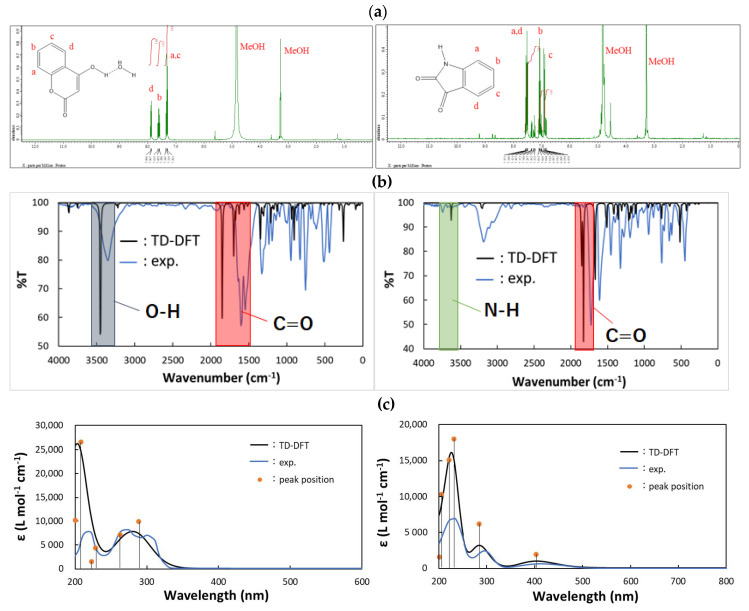
Comparison with experimental and DFT computational results of (**a**) NMR (only experimental) (**b**) IR and (**c**) UV-vis spectra for (**left**) **CA1** and (**right**) **CA4**.

**Figure 3 molecules-25-05691-f003:**
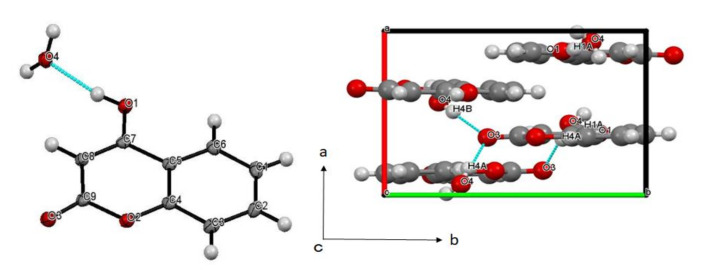
(**left**) Molecular and (**right**) crystal structure of **CA1**. Selected bond distances [Å] and angles [°]: O1-C7 = 1.3176(16), O2-C9 = 1.3720(15), O2-C4 = 1.3774(16), O3-C9 = 1.2341(17), C7-C8 = 1.3712(18), C8-C9 = 1.4206(17), O1-C7-C8 = 124.65(12), O1-C7-C5 = 115.78(11), C9-O2-C4 = 121.05(10), O2-C9-C8 = 119.29(11), O3-C9-O2 = 114.55(11), O3-C9-C8 = 126.15(12).

**Figure 4 molecules-25-05691-f004:**
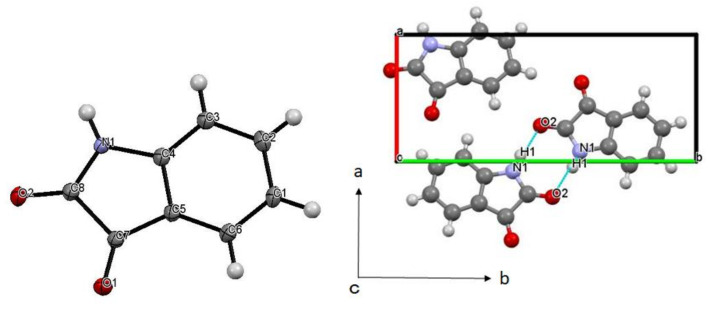
(**left**) Molecular and (**right**) crystal structure of **CA4**. Selected bond distances [Å] and angles [°]: O1-C7 = 1.2113(16), O2-C8 = 1.2215(16), N1-C8 = 1.3544(17), N1-C4 = 1.4102(16), C5-C7 = 1.4658(17), C7-C8 = 1.5670(18), C8-N1-C4 = 111.22(11), C3-C4-N1 = 127.32(12), C5-C4-N1 = 111.04(11), O1-C7-C5 = 131.17(12), O2-C8-C7 = 126.03(12), N1-C8-C7 = 105.95(10).

**Figure 5 molecules-25-05691-f005:**
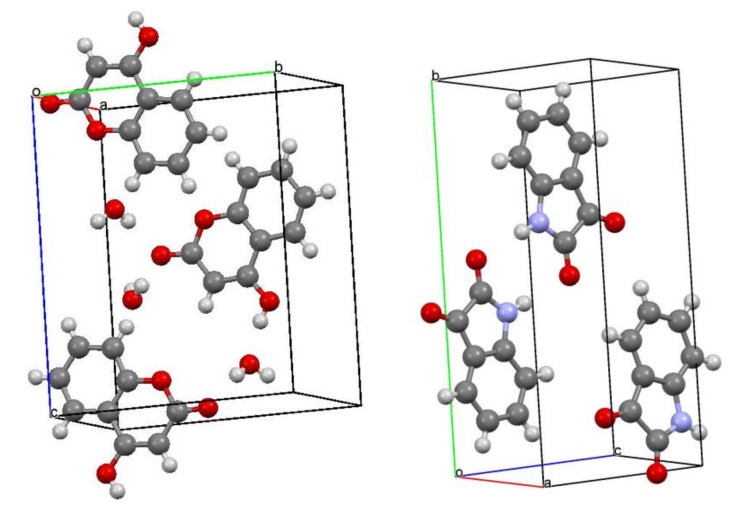
Crystal packings of (**left**) **CA1** and (**right**) **CA4**.

**Table 1 molecules-25-05691-t001:** Crystallographic data of **CA1** and **CA4**.

	CA1 CCDC 2043410	CA4 CCDC 2043411
Empirical formula	C_9_H_6_O_3_·H_2_O	C_8_H_5_NO_2_
Formula weight	180.16	147.13
Crystal system	Orthorhombic	Monoclinic
Space group	*P*2_1_2_1_2_1_ (# 19)	*P*2_1_/*c* (# 14)
Z	4	4
*a*(Å)	6.762(13)	6.1473(3)
*b*(Å)	9.943(19)	14.5760(6)
*c*(Å)	12.207(2)	7.0495(3)
*β*(˚)		93.744(7)
*V*(Å^3^)	820.71(3)	630.31(4)
*ρ_calc_*(g/cm^3^)	1.458	1.550
*μ*(mm^−1^)	0.116	0.114
*F*(0 0 0)	376	304.00
Goodness of fit	1.089	1.070
*R_1_*[*I > 2σ(I)*]	0.0304	0.0422
*wR_2_*	0.0808	0.1153

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
