# Peer review of "Success or Failure of Chiral Crystallization of Similar Heterocyclic Compounds"

_molecules, 2020, doi:10.3390/molecules25235691_

Round 1

Reviewer 1 Report

In the manuscript, the authors accidentally obtained single crystals of 4-Hydroxycoumarins and isatin in the synthesis of hydrazone-Schiff base ligand by reacting 4-hydroxycoumarin or isatin with pyrazinamide in the presence of glacial acetic acid in ethanolic solution. The target ligand was not synthesized, instead of the recovery of the starting materials which yielded the crystal suitable for single crystal X-ray diffraction study. It is not surprising that single crystals of 4-Hydroxycoumarins and isatin showed some difference. The result suggested that the alignment of the planar cyclic sites slightly tilted from parallel in the crystal packing led to the chiral crystallization in 4-Hydroxycoumarin monohydrate. The manuscript may be published after the following points are addressed.

  • How about the single crystal directly obtained by the crystallization of 4-Hydroxycoumarin or isatin from ethanol, is it different or same?
  • Why was the different refluxing temperature (90 OC verus 70 OC) required in the synthesis of CA1 and CA4? Same solvent, different refluxing temperature?
  • CA1 was obtained as yellow crystals in Section 2.1, while a colorless (CA1) for X-ray analysis in Section 2.4?
  • There are some errors in the manuscript, a careful check should be performed.

Reviewer 2 Report

Planar structures of analytes should be presented in the manuscript.

The Reference section should be revised, it is not written according to journal recommandation and requirements

The importance of chirality should be emphasized in the introduction section.

The aim of the study is not clearly presented and described in the Introduction section.

Conclusion section must be rewritten, the aim and the results should be correlated

Reviewer 3 Report

This is interesting work with great degree of scientific potential. However, the authors could not present the work in the light of most interesting aspect as the relevance of this results for the formation of prebiotic homochirality. Furthermore, I would suggest to describe more about the chirality of the chiral crystals in terms of helicity (?) and its reproducibility; less expensive powder-crystallography can be used. A similar work was described by the group: Ueki, H.; Soloshonok, V. A. New Sterically Driven Mode for Generation of Helical Chirality, Org. Lett. 2009, 11, 1797-1800. The authors can pick up some ideas on the presentation and importance of these results. Also, the authors should pay attention to the terminology; "chiral crystallization" is absolutely incorrect term as crystallization cannot be chiral. 

Round 2

Reviewer 1 Report

The authors have revised the manuscript according to the reviewers' comments, I therefore recommend it for the publication in Molecules.

Reviewer 2 Report

The article can be published in the current form